# Capacitorless One-Transistor Dynamic Random-Access Memory with Novel Mechanism: Self-Refreshing

**DOI:** 10.3390/nano14020179

**Published:** 2024-01-12

**Authors:** Sang Ho Lee, Jin Park, Young Jun Yoon, In Man Kang

**Affiliations:** 1School of Electronic and Electrical Engineering, Kyungpook National University, Daegu 41566, Republic of Korea; jim782@knu.ac.kr (S.H.L.); jdefs12@knu.ac.kr (J.P.); 2Department of Electronic Engineering, Andong National University, 1375 Gyengdong-ro, Andong 36729, Republic of Korea

**Keywords:** junctionless field-effect transistor, silicon-on-insulator, one-transistor dynamic random-access memory, self-refreshing operation

## Abstract

In this paper, we propose for the first time a self-refreshing mechanism in a junctionless field-effect transistor (JLFET) based on one-transistor dynamic random-access memory (1T-DRAM) with a silicon-on-insulator (SOI) structure. The self-refreshing mechanism continuously creates holes by appropriately generating impact ionization during the holding process through the application of an appropriate operation bias voltage. This leads to self-refreshing, which prevents the recombination of holes. When using the self-refreshing mechanism for the proposed device, the sensing margins were 15.4 and 12.7 μA/μm at 300 and 358 K, respectively. Moreover, the device achieved an excellent performance retention time of >500 ms, regardless of the temperature of the 1T-DRAM with a single gate. Furthermore, cell disturbance analysis and voltage optimization were performed to evaluate the in-cell reliability of the proposed device. It also showed excellent performance in terms of energy consumption and writing speed.

## 1. Introduction

Dynamic random-access memory (DRAM) is one of the most important devices in electronic systems today. Many researchers have constantly attempted to shrink its size to accommodate more devices on the same chip size. However, it has reached its scaling limit. Thus, some researchers have proposed one-transistor dynamic random-access memory (1T-DRAM) without a capacitor instead of conventional DRAM [1,2,3,4,5,6]. Because 1T-DRAM uses the principle of the floating body effect, the capacitor can be eliminated. 1T-DRAM is easy to manufacture and has excellent logic device compatibility. However, a longstanding weakness of 1T-DRAM is its short retention time (RT). Various structures such as nanotube-based 1T-DRAM, Poly-Si metal-oxide-semiconductor field-effect transistor (MOSFET)-based dual-gate 1T-DRAM, three-dimensional (3D) stacked Poly-Si MOSFET-based 1T-DRAM, vertical double-gate 1T-DRAM, IGZO-based 1T-DRAMs, and InGaAs-based 1T-DRAMs have been proposed to improve the memory retention characteristics [7,8,9,10,11,12,13,14,15]. However, as mentioned earlier, the structures and their fabrication processes are complex. In addition, it is expensive to fabricate a memory device, and even if the device is designed using a complicated fabrication method, the RT of the device does not meet expectations. 1T-DRAMs with two or more gates can achieve a long RT [16,17,18,19,20]. However, in such cases, the probability of a disturbance error is higher than in single-gate 1T-DRAM devices [21]. Herein, 1T-DRAM is based on a conventional silicon-on-insulator (SOI) fin field-effect transistor (FinFET) structure. This 1T-DRAM is superior to other 1T-DRAMs in terms of ease of fabrication. In addition, it uses a novel self-refreshing mechanism. We designed and optimized JLFET-based 1T-DRAM with a long RT of >500 ms. Furthermore, the array improves reliability by optimizing the operation bias and accounting for bit-line (BL) and word-line (WL) disturbances. A technology computer-aided design (TCAD) simulation based on Sentaurus was used to investigate the proposed 1T-DRAM [22].

## 2. Device Structure and Simulation Methodology

Figure 1 shows a three-dimensional (3D) view and schematic cross-section of the JL SOI-FinFET-based 1T-DRAM. The gate length (*L*_g_), the source/drain length (*L*_s_, *L*_d_), the height of the fin (*H*_fin_), and the fin width (*W*_fin_) are 100, 50, 50, and 30 nm, respectively. The source, channel, drain, and substrate region are made of single crystalline silicon. To meet the requirement of 0.3 nm equivalent oxide thickness stipulated in the technology roadmap for 2025 [23], we designed the gate dielectric (HfO_2_) thickness (*T*_ox_) as 3 nm. The buried oxide is made of silicon oxide (SiO_2_). The gate work function is 5.0 eV to deplete the body region. The doping concentrations of the source, channel, and drain regions are 3 × 10^18^ cm^−3^ (*n*-type). We referred to the device parameters of the proposed 1T-DRAM in [16]. We designed a channel length of 100 nm considering that a long channel length improves storage ability because the size of the storage area is related to the channel length of the proposed 1T-DRAM [15]. Table 1 summarizes each device parameter for the proposed device. Sentaurus TCAD simulation was used to investigate the transfer characteristics and memory performances. For accurate TCAD simulation, various physical models such as the Auger recombination model, Shockley–Read–Hall recombination model, doping-dependent and field-dependent mobility models, Fermi–Dirac statistical model, nonlocal band-to-band tunneling model, trap-assisted-tunneling model, bandgap narrowing model, and quantum confinement model were considered [22].

## 3. Results and Discussion

### 3.1. Characteristics of 1T-DRAM Cell

Figure 2a,b show *I*_d_–*V*_g_ the transfer and output curves of the proposed 1T-DRAM, respectively, at 300 K. As shown in Figure 2a, the 1T-DRAM exhibits a switching operation similar to conventional junctionless transistors when the voltage is low. The JLFET operates without traditional p–n junctions and instead relies on a uniformly doped semiconductor channel. In a JLFET, a gate with a high work function generally depletes the channel region, allowing uniform control of the current flow between the source and drain [24]. When the 1T-DRAM exceeds a specific drain voltage, impact ionization occurs, making the transfer curve slope considerably steep. This phenomenon does not occur at low drain voltages. The drain voltage is gradually increased, causing impact ionization [25,26,27]. The *I*_d_–*V*_d_ output curve shown in Figure 2b demonstrates that the occurrence of impact ionization is dependent on the gate voltage. The aforementioned figures show that the occurrence of impact ionization is dependent on the gate and drain voltages, and by optimizing these voltages, we can design 1T-DRAM that exhibits a self-refreshing phenomenon in a hold state.

Figure 3 shows the transient memory characteristics of 1T-DRAM. For program operation, the gate and drain voltages were applied at 0.2 and 2.5 V, respectively. In this case, strong impact ionization occurs, generating numerous electron–hole pairs. This is shown in Figure 4 as the hole impact ionization rate. In the hold ‘1’ state, strong impact ionization occurs at the junction of the channel and drain due to the substantial drain voltage. Consequently, the hole density in hold ‘1’ increases, resulting in state ‘1’. Conversely, in hold ‘0’, the gate voltage and drain voltage are 0.2 and −0.5 V, respectively, resulting in impact ionization that rarely occurs compared with write ‘1’. The operation bias is summarized in Table 2.

Figure 5a,b show the results of the read currents and sensing margins depending on the drain voltage at 300 K. In Figure 5a, when the drain voltage is 0.0, 0.1, and 0.2 V, the hold time for state ‘1’ increases, the number of excess holes decreases and the drain current decreases. This is because recombination occurs actively, resulting in the excess hole recombination rate (R_p_) becoming more substantial than the excess hole recombination rate (G_p_). The drain voltage of 0.4 V shows that as the hold time for state ‘0’ increases, the number of excess holes increases and the read ‘0’ increases. This phenomenon occurs when R_p_ becomes lesser than G_p_ and the number of recombination holes becomes lesser than the amount of self-refreshing holes. Thus, the sensing margin and RT decrease at a drain voltage of 0.4 V. At 300 K, the appropriate drain voltage is 0.3–0.35 V. Regarding the RT, an excellent performance RT of >1 s was achieved in the case of 1T-DRAM with a single gate. When using a self-refreshing mechanism, optimizing the aforementioned drain voltage is critical; otherwise, the hold state ‘1’ or ‘0’ may be disturbed.

Figure 6a,b show the results of the read current and sensing margin depending on the drain voltage at 358 K. Figure 6a shows that R_p_ becomes greater than G_p_ when the drain voltage is 0.0, 0.1, 0.2, and 0.3 V. At drain voltage of 0.4 V, R_p_ is lesser than G_p_. The rates of R_p_ and G_p_ must be in equilibrium to maintain a longer RT. However, except at 0.35 V, there is an imbalance between the recombination and generation rates. Therefore, the sensing margins cannot be maintained for a long time. At 0.35 V, the recombination and generation rates are in equilibrium. Figure 6b shows that the sensing margin maintains the initial value for 1 s or even more. Regarding the RT, an excellent performance of RT of >1 s was achieved even at high temperatures.

### 3.2. Array Characteristics of 1T-DRAM and Optimization of Operation Bias

A 1T-DRAM cell array comprises multiple independent memory devices and operates with a shared WL and BL. Therefore, disturbance by a WL or BL is also one of the most critical issues related to the 1T-DRAM circuit [21,28,29,30,31,32,33,34]. Figure 7 shows a circuit diagram of the proposed 1T-DRAM array considering the disturbance error. This array comprises a WL and BL. The cell is selected and controlled by applying a bias in each line. The disturbance errors are analyzed depending on the conditions of disturbance that are set for the programs, erases, and read operations. The results of the disturbance errors are summarized in Table 3. The bias condition of the 1T-DRAM directly affects the disturbance error. The disturbance phenomenon of the 1T-DRAM is primarily influenced by the erase and program biases of the BL and WL. Because the positive bias of the WL slightly reduces the stored holes under G2, the read current at the ‘1’ state decreases.

However, excluding the aforementioned requirements, the remaining disturbance conditions are close to 100%, which can be considered reliable in the remaining states. Therefore, we must optimize the bias condition when operating during the conditions specified in Table 3 (highlighted in red). The bias condition before the optimization is the same as the value in Table 3.

First, the gate and drain voltage were optimized to resolve disturbance errors during the program operation. When the WL and BL were 0.2 and 2.5 V, respectively, the disturbance values were all functionless (labeled in red). We lowered the WL 0.1 and 0.0 V to minimize the influence of the disturbance. Regarding the BL, the voltage was increased from 2.5 to 4.5 V to strengthen the write operation. Consequently, the disturbed and undisturbed conditions of the proposed 1T-DRAM improved from 50.0% and 45.8% to 87.9% and 83.3%, respectively. When the WL was 0.0 V, the impact ionization during writing was attenuated and the sensing margin became an error, as shown in Table 4 (highlighted in orange). The disturbed and undisturbed conditions for the various biases are shown in Table 4.

Second, to minimize the disturbance error of the erase operation, optimizing the gate and drain voltages of the erase operation is imperative. Table 5 shows the sensing margin percentage between disturbed and undisturbed conditions for the various bias operations at write ‘0’ under operation temperatures of 300 and 358 K. When the WL was 0.20 and 0.15 V and the BL was −0.50 and −0.40 V, the disturbed and undisturbed conditions at the hold ‘1’ erase operation were functionless (labeled in red). During the erase operation, a voltage of 0.20 V was previously applied to the WL, but it was lowered to 0.15 V to minimize the effect of the disturbance. During the erase operation, not only the WL but also the BL was optimized. The simulation was conducted by varying the voltage from −0.50 to −0.30 V. When the BL voltage was changed from −0.50 to −0.40 V, the disturbance error was reduced, but some cases were still functionless. When the BL was changed to −0.35 V, it showed good disturbance immunity. When the BL was applied to −0.30 V, the holes gathered in the body region could not be swept out during the erase operation. It could not perform 1T-DRAM memory operations. This appears to be a sensing margin error. When we optimized the erase bias condition, it showed good immunity to disturbance when operating at 300 and 358 K. The disturbed and undisturbed conditions of the proposed 1T-DRAM improved to 71.0 and 66.2%, respectively. Thus, considering all disturbance errors, the erase and program operating voltages were −0.35 and 4.5 V for the BL and 0.15 and 0.1 V for the WL.

Figure 8a,b show the sensing margin and RT when an operating voltage that minimizes the disturbance error is applied. The optimized operation bias of gate and drain voltages are 0.1 and 4.5 V in write, 0.15 and −0.35 V in erase, 0.1 and 0.5 V in hold, and −0.2 and 0.35 V in the read operations, respectively. In Figure 8a, the read ‘1’ current does not decrease but remains constant. This means that during hold ‘1’, the impact ionization phenomenon that prevents recombination continues to occur, resulting in a self-refreshing mechanism. In Figure 8b, when the hold time approaches 500 ms, a hole is generated due to its tendency to return to equilibrium in the body region at hold ‘0’, and although the sensing margin is slightly reduced, there is little disruption to the RT. Therefore, the sensing margin is constant for a longer time without an additional writing operation. Figure 8b shows that the sensing margin is constant. This is sufficient for the RT to meet the requirement of 64 ms, the memory criterion set by the international roadmap for devices and systems (>64 ms), even for devices at voltages contributing to the disturbance error [23].

### 3.3. Memory Characteristics and Energy Consumption of 1T-DRAM

Table 6 compares the proposed 1T-DRAM and the recently reported 1T-DRAM devices regarding the sensing margin, retention time, features, and challenges. Regarding the sensing margin, the proposed 1T DRAM was superior to the others (excluding the Poly-Si nanotube 1T-DRAM). The proposed 1T-DRAM was superior or mainly equivalent to the others regarding the retention time (excluding IGZO-based 1T-DRAM). However, except for the proposed 1T-DRAM, the other 1T-DRAMs have several challenges. For example, the Poly-Si MOSFET, Poly-Si nanotube 1T-DRAM, and 3D stacked 1T-DRAM have complex fabrication challenges because of their complex structures [8,9,16]. The vertical double gate 1T-DRAM is compatible with the CMOS process but shows poor memory characteristics [10]. The IGZO-based 1T-DRAM and InGaAs-based 1T-DRAM show superior retention times. However, those 1T-DRAMS are not CMOS compatible [11,14]. 1T-FEMOS shows superior retention time and read window. However, it shows a poor absolute sensing margin (I_R1_-I_R0_) [35]. Consequently, the proposed 1T-DRAM shows excellent memory characteristics, can also be CMOS-compatible, and has an easy fabrication process. In addition, the 1T-DRAM cell exhibited a fast write speed of 10 ns during the write, hold, and read operations. Therefore, these results show the potential of 1T-DRAM for high-speed memory as a substitute for a conventional 1T-1C DRAM because the write speed of the write time of DRAM is typically measured in tens of nanoseconds [27].

Energy consumption is a crucial metric for memory devices. Table 7 shows the energy consumption calculated by multiplying |V_D_|, I_D_, and the operation time for each operation of the proposed 1T-DRAM [27]. The program (write ‘1’) and erase (write ‘0’) operations consumed 411.3 fJ/bit and 17.5 fJ/bit, respectively. The ‘1’ and ‘0’ read operations consumed 8.5fJ/bit and 0.006fJ/bit, respectively. The proposed 1T-DRAM used a self-refreshing mechanism but only consumed 1.8 fJ/bit and ~0 fJ/bit at holds ‘1’ and ‘0’, respectively. The energy consumption of the proposed 1T-DRAM, conventional 1T-1C DRAM, and the recently reported 1T-DRAM are compared in Table 8. It showed 22 times lower energy consumption than the conventional 1T-1C DRAM. The proposed 1T-DRAM was comparable to other 1T-DRAM devices, although it used impact ionization.

## 4. Conclusions

Herein, we propose for the first time a self-refreshing mechanism for JLFET-based 1T-DRAM with an SOI structure. The self-refreshing mechanism continuously creates holes by appropriately generating impact ionization during the holding process through application of an optimized operating bias voltage. This causes self-refreshing, which prevents recombination of holes. When the self-refreshing method was used for the proposed device, the sensing margins were 15.4 and 12.7 μA/μm at 300 and 358 K, respectively. Moreover, the device achieved an excellent performance RT of >500 ms regardless of the temperature in the 1T-DRAM with a single gate. Additionally, cell disturbance analysis and voltage optimization were performed to evaluate the in-cell reliability of the proposed device. It also showed excellent performance in terms of energy consumption and write speed.

## Figures and Tables

**Figure 1 nanomaterials-14-00179-f001:**
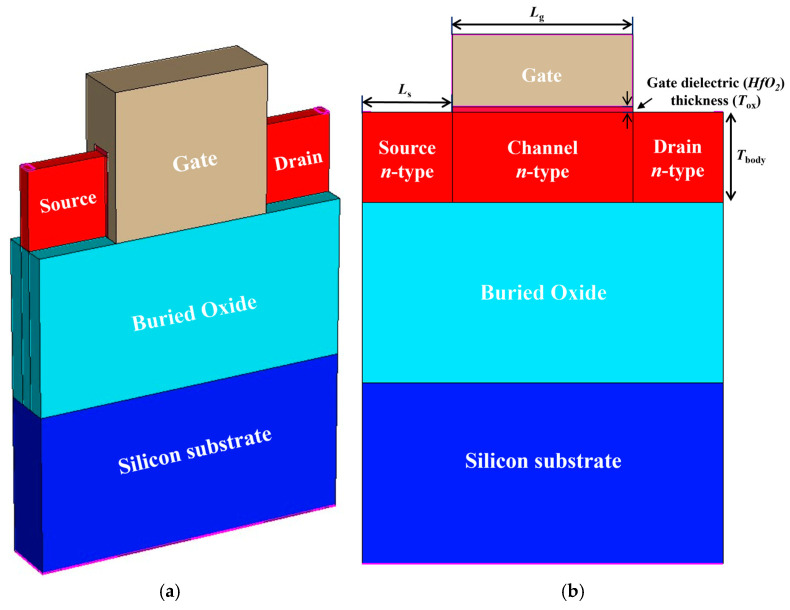
(**a**) Three-dimensional schematic of the proposed JL SOI-FinFET-based 1T-DRAM and (**b**) cross-sectional view.

**Figure 2 nanomaterials-14-00179-f002:**
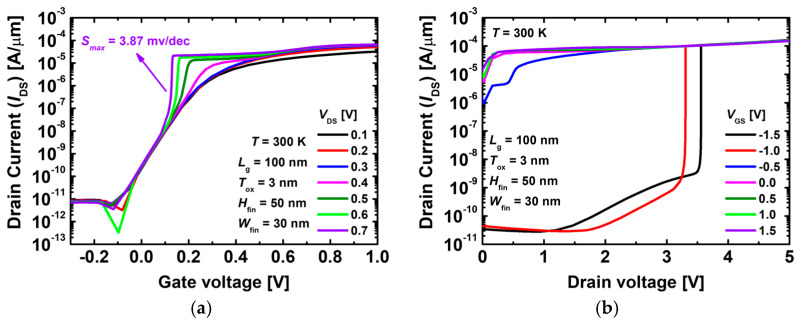
(**a**) *I*_d_–*V*_g_ transfer curve and (**b**) *I*_d_–*V*_d_ output curve of the proposed 1T-DRAM.

**Figure 3 nanomaterials-14-00179-f003:**
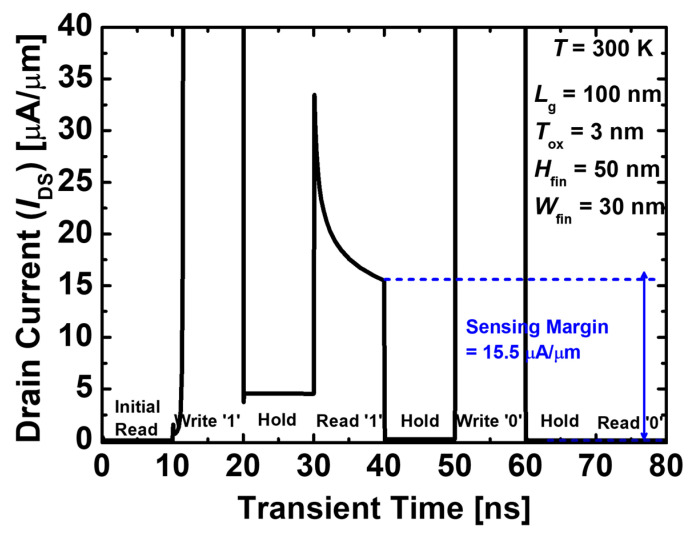
Transient characteristic of the proposed 1T-DRAM. The operating time is 10 ns.

**Figure 4 nanomaterials-14-00179-f004:**
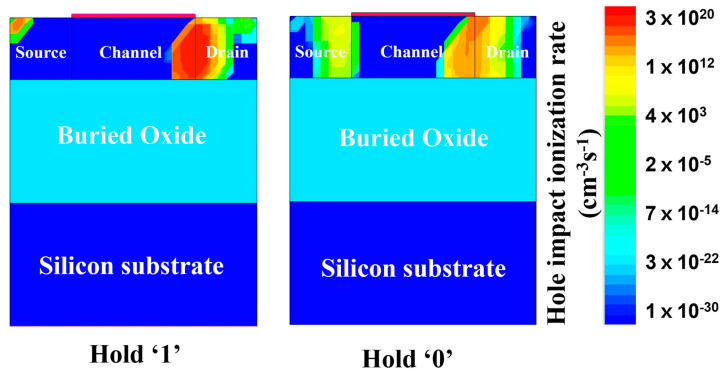
Contour map of the hole impact ionization rate of the proposed 1T-DRAM cell in states ‘1’ and ‘0’.

**Figure 5 nanomaterials-14-00179-f005:**
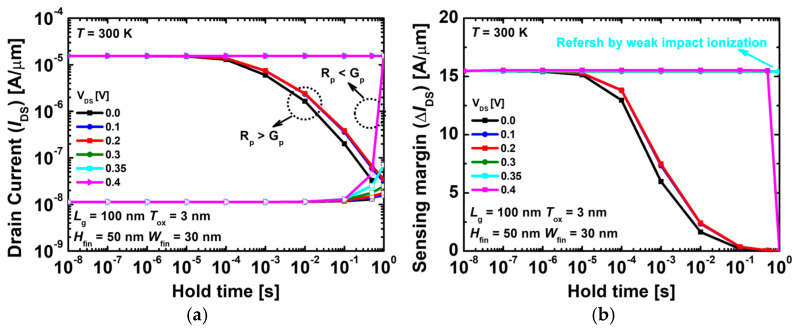
(**a**) Read ‘1’ and ‘0’ currents and (**b**) sensing margins as a function of the hold time depending on the drain voltages at 300 K.

**Figure 6 nanomaterials-14-00179-f006:**
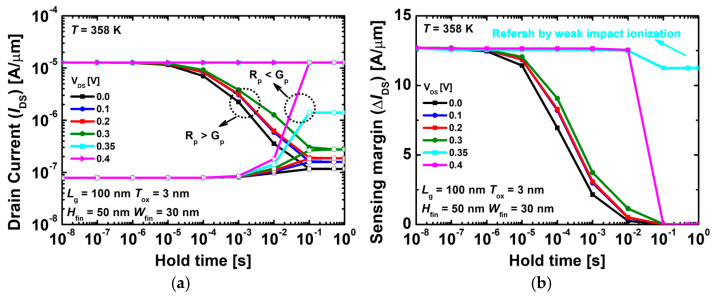
(**a**) Read ‘1’ and ‘0’ currents and (**b**) sensing margins as a function of the hold time with drain voltages at 358 K.

**Figure 7 nanomaterials-14-00179-f007:**
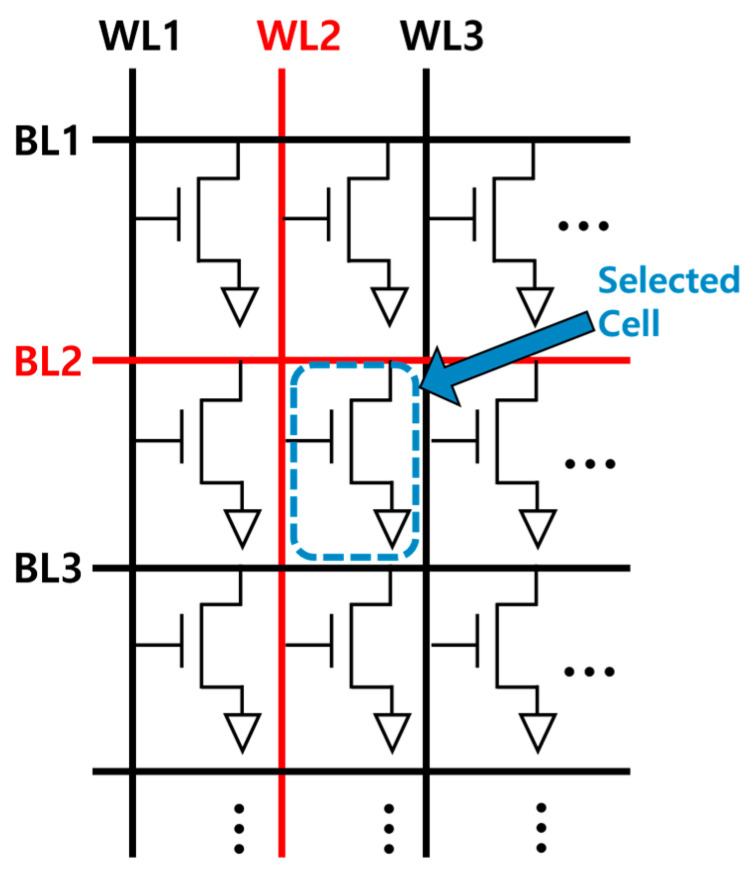
The proposed 1T-DRAM cell array and disturbances among neighboring cells.

**Figure 8 nanomaterials-14-00179-f008:**
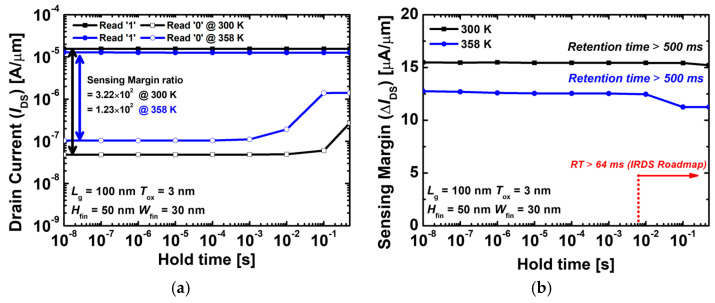
(**a**) Sensing margin and (**b**) retention time of the proposed 1T-DRAM with an optimized bias voltage. The optimized operation bias of gate and drain voltage are 0.1 and 4.5 V in write, 0.15 and −0.35 V in erase, 0.1 and 0.5 V in read, and −0.2 and 0.35 V in hold operation.

**Table 1 nanomaterials-14-00179-t001:** Geometric parameters of the proposed 1T-DRAM used for simulation.

Parameters	Values
Gate length (*L*_g_)	100 nm
Source/Drain length (*L*_s_, *L*_d_)	50 nm
Gate dielectric (HfO_2_) thickness (*T*_ox_)	3 nm
Fin height (*H*_fin_)	50 nm
Fin width (*W*_fin_)	30 nm
Source/Body/Drain doping concentration	*n*-type, 3 × 10^18^ cm^−3^
Gate metal work function	5.0 eV

**Table 2 nanomaterials-14-00179-t002:** Operating bias scheme of the proposed 1T-DRAM for memory performance.

Operation	Program (Write ‘1’)	Erase (Write ‘0’)	Read	Hold
Gate voltage [V]	0.2	0.2	0.1	−0.2
Drain voltage [V]	2.5	−0.5	0.5	0.35

**Table 3 nanomaterials-14-00179-t003:** Sensing margin percentage (%) between disturbed and undisturbed conditions under operation temperatures of 300 and 358 K.

Disturbance Conditions			SM_disturbed_/SM_undisturbed_ [%] @ t_disturbance_ = 10 ns	
Device Condition	Disturb Source	Device Operation	300 K	358 K
Hold ‘1’	Bit-line	Erase	17.3%	15.2%
		Program	100.3%	100.4%
		Read	100.2%	100.5%
	Word-Line	Erase	50.0%	45.8%
		Program	50.0%	45.8%
		Read	87.7%	83.6%
Hold ‘0’	Bit-line	Erase	100.0%	100.0%
		Program	100.0%	100.0%
		Read	100.0%	100.0%
	Word-Line	Erase	100.0%	100.0%
		Program	100.0%	100.0%
		Read	100.0%	100.0%

**Table 4 nanomaterials-14-00179-t004:** Sensing margin percentage (%) between disturbed and undisturbed conditions for the various bias operations at write ‘1’ under operation temperatures of 300 K and 358 K.

Disturbance Conditions			SM_disturbed_/SM_undisturbed_ [%] @ t_disturbance_ = 10 ns									
			*V*_gs_ = 0.2 V, *V*_ds_ = 2.5 V		*V*_gs_ = 0.2 V, *V*_ds_ = 4.5 V			*V*_gs_ = 0.1 V, *V*_ds_ = 4.5 V			*V*_gs_ = 0.0 V, *V*_ds_ = 4.5 V	
Device Condition	Disturb Source	Device Operation	300 K	358 K	300 K		358 K	300 K		358 K	300 K	358 K
Hold ‘1’	Bit-line	Erase	17.3%	15.2%	17.3%		15.1%	17.3%		15.1%	-	-
		Program	100.3%	100.4%	99.9%		99.9%	100.0%		99.9%	-	-
		Read	100.2%	100.5%	100.0%		100.0%	100.1%		100.0%	-	-
	Word-Line	Erase	50.0%	45.8%	50.0%		45.8%	50.1%		45.8%	-	-
		Program	50.0%	45.8%	50.0%		45.8%	87.9%		83.3%	-	-
		Read	87.7%	83.6%	87.7%		83.3%	87.9%		83.3%	-	-
Hold ‘0’	Bit-line	Erase	100.0%	100.0%	100.0%		100.0%	100.0%		100.0%	-	-
		Program	100.0%	100.0%	99.9%		99.3%	99.9%		99.3%	-	-
		Read	100.0%	100.0%	100.0%		100.0%	100.0%		100.0%	-	-
	Word-Line	Erase	100.0%	100.0%	100.0%		100.0%	100.0%		100.0%	-	-
		Program	100.0%	100.0%	100.0%		100.0%	100.0%		100.0%	-	-
		Read	100.0%	100.0%	100.0%		100.0%	100.0%		100.0%	-	-
Operation temp	300 K		Percentage (%)	90~100%		60~90%			0~60%		SM Error	
	358 K											

**Table 5 nanomaterials-14-00179-t005:** Sensing margin percentage (%) between disturbed and undisturbed conditions for the various bias operations at write ‘0’ under operation temperatures of 300 K and 358 K.

Disturbance Conditions			SM_disturbed_/SM_undisturbed_ [%] @ t_disturbance_ = 10 ns									
			*V*_gs_ = 0.20 V, *V*_ds_ = −0.50 V		*V*_gs_ = 0.15 V, *V*_ds_ = −0.50 V		*V*_gs_ = 0.15 V, *V*_ds_ = −0.40 V		*V*_gs_ = 0.15 V, *V*_ds_ = −0.35 V		*V*_gs_ = 0.15 V, *V*_ds_ = −0.30 V	
Device Condition	Disturb Source	Device Operation	300 K	358 K	300 K	358 K	300 K	358 K	300 K	358 K	300 K	358 K
Hold ‘1’	Bit-line	Erase	17.3%	15.1%	17.3%	15.1%	49.9%	45.7%	71.0%	66.2%	-	-
		Program	100.0%	99.9%	100.0%	99.9%	100.0%	99.9%	100.0%	99.9%	-	-
		Read	100.1%	100.0%	100.1%	100.0%	100.1%	100.0%	100.1%	100.0%	-	-
	Word-Line	Erase	50.1%	45.8%	71.0%	66.3%	71.0%	66.2%	71.0%	66.2%	-	-
		Program	87.9%	83.3%	87.9%	83.3%	87.8%	83.3%	87.8%	83.3%	-	-
		Read	87.9%	83.3%	87.9%	83.3%	87.8%	83.3%	87.8%	83.3%	-	-
Hold ‘0’	Bit-line	Erase	100.0%	100.0%	100.0%	100.0%	100.0%	100.0%	100.0%	100.0%	-	-
		Program	99.9%	99.3%	99.9%	99.3%	99.8%	99.3%	99.6%	98.9%	-	-
		Read	100.0%	100.0%	100.0%	100.0%	100.0%	100.0%	100.0%	100.0%	-	-
	Word-Line	Erase	100.0%	100.0%	100.0%	100.0%	100.0%	100.0%	100.0%	100.0%	-	-
		Program	100.0%	100.0%	100.0%	100.0%	100.0%	100.0%	100.0%	100.0%	-	-
		Read	100.0%	100.0%	100.0%	100.0%	100.0%	100.0%	100.0%	100.0%	-	-
Operation temp	300 K		Percentage (%)		90~100%		60~90%		0~60%		SM Error	
	358 K											

**Table 6 nanomaterials-14-00179-t006:** Comparison between the proposed 1T-DRAM and recently published 1T-DRAMs.

Device	Reference	Sensing Margin (μA/μm)	Retention Time	Features and Challenges
This work	-	12.7~15.4	>500 ms	CMOS compatible
Poly-Si MOSFET 1T-DRAM	[8]	8.7	704 ms	A complex fabrication process of dual gates
Poly-Si nanotube 1T-DRAM	[9]	422	120 ms	A complex fabrication process of the inner and outer gate
Vertical Double-gate 1T-DRAM	[10]	3~6	25 ms	Poor memory characteristics
IGZO-based 1T-DRAM	[11]	-	>400 s	CMOS incompatible
InGaAs-based 1T-DRAM	[14]	~2	>1 ms	CMOS incompatible
3D stacked 1T-DRAM	[16]	17.4	200 ms	Complex fabrication process
1T-FeMOS	[35]	~1	5 s	Good retention time but poor sensing margin

**Table 7 nanomaterials-14-00179-t007:** Energy consumption of the proposed 1T-DRAM in respect to operation.

Operation	Program (Write ‘1’)	Erase (Write ‘0’)	Read	Hold
Gate voltage [V]	0.1	0.15	0.1	−0.2
Drain voltage [V]	4.5	−0.35	0.5	0.35
Drain current [A]	9.1395 × 10^−6^	4.9959 × 10^−6^	R ‘1’: 1.7039 × 10^−6^ R ‘0’: 1.2374 × 10^−9^	H ‘1’: 5.0051 × 10^−7^ H ‘0’: 1.7928 × 10^−14^
Operation time [ns]	10	10	10	10
Energy consumption [J] (E = V_D_ × I_D_ × Time)	411.3 fJ	17.5 fJ	8.5 fJ 0.006 fJ	1.8 fJ 0 fJ

**Table 8 nanomaterials-14-00179-t008:** Comparison between the proposed 1T-DRAM, the conventional 1T-1C DRAM, and the recently reported 1T-DRAMs regarding energy consumption.

Device	Energy Consumption (E = V_D_ × I_D_ × Time)
This work	439 fJ
Conventional 1T-1C DRAM [36]	>10,000 fJ
SiGe QW 1T-DRAM [37]	383 fJ
Z^2^-FET [38]	1000~4000 fJ

## Data Availability

Data are contained within the article.

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
