# Peer review of "Capacitorless One-Transistor Dynamic Random-Access Memory with Novel Mechanism: Self-Refreshing"

_nanomaterials, 2024, doi:10.3390/nano14020179_

Round 1
Reviewer 1 Report
Comments and Suggestions for Authors
a) One-transistor dynamic random-access memory without capacitor has been studied a lot in recent years. In this study, the authors studied 1T-DRAM based on conventional SOI FinFET structure. There was no structural innovation, so the “Novel Capacitorless 1T DRAM” is not particularly appropriate.
b) The authors create holes by generating impact ionization by applying an operating bias voltage. This will definitely affect the read and write life, is there any relevant assessment?
Comments on the Quality of English LanguageModerate editing of English language required
Reviewer 2 Report
Comments and Suggestions for Authors
The SOI based 1T DRAM has been proposed for decades.
One basic issue preventing in real application, compared with existing 1T-1C DRAM, is the power efficiency to generate holes and change the Vth. Please compare the write/erase power with commercial 1T-1C DRAM.
Also compare the write/erase time with commercial 1T-1C DRAM.
Please write your responses and potential solutions in both abstract and introduction.
Reviewer 3 Report
Comments and Suggestions for Authors
In this manuscript, the authors have proposed a self-refreshing mechanism in a junctionless field-effect transistor (JLFET) based on one-transistor dynamic random-access memory (1T-DRAM) with a silicon-on-insulator (SOI) structure. The authors stated that the self-refreshing mechanism continuously creates holes by appropriately generating impact ionization during the holding process through application of an appropriate operation bias voltage. Then, the auhors found that when using the self-refreshing mechanism for the proposed device, the sensing margins were 15.4 and 12.7 μA/μm at 300 and 358 K, respectively. Moreover, the device achieved excellent performance retention time of >500 ms regardless of the temperature in the 1T-DRAM with a single gate. Furthermore, cell disturbance analysis and voltage optimization were performed to evaluate the in-cell reliability of the proposed device.The topic is interesting for understanding the DRAM and improving the property, and then developing high-performance DRAM. However, several questions should be addressed before this paper can be considered to publish in nanomaterials.
1.In this manuscript, the authors only described the structure of the JL SOI-Fin-FET-based 1T-DRAM, but lack of material characterization
2.What are the advantages of JL SOI-Fin-FET-based 1T-DRAM over current IGZO-based 2T0C DRAM or 1T1C DRAM? The authors should provide some relevant data for comparison.
3.In addition to the characterization of some of the properties in the manuscript, it is recommended that the authors test some other properties such as power consumption, storage density, retention characteristics, etc.
4.The authors are suggested to cite more relevant references to better describe the importance of this work, such as [First Demonstration of Dual-Gate IGZO 2T0C DRAM with Novel Read Operation, One Bit Line in Single Cell, ION=1500 μA/μm@VDS=1V and Retention Time>300s], [Nature Communications, 5 : 4598 (2014)].
Comments on the Quality of English LanguageNo problem with the language
Round 2
Reviewer 2 Report
Comments and Suggestions for Authors
The authors have answered all my previous comments on energy efficiency and retention time. This paper is important for DRAM area. It will be even better if the authors can compare this device with Ferroelectric DRAM:
C. H. Cheng and A. Chin, “Low-Leakage-Current DRAM-like Memory Using a One-Transistor Ferroelectric MOSFET with a Hf-based Gate Dielectric,” IEEE Electron Device Lett., vol. 35, pp. 138-140, Jan. 2014.
